# A Hybrid Preprocessor DE-ABC for Efficient Skin-Lesion Segmentation with Improved Contrast

**DOI:** 10.3390/diagnostics12112625

**Published:** 2022-10-29

**Authors:** Shairyar Malik, Tallha Akram, Imran Ashraf, Muhammad Rafiullah, Mukhtar Ullah, Jawad Tanveer

**Affiliations:** 1Department of Electrical and Computer Engineering, Wah Campus, COMSATS University Islamabad, G.T. Road, Wah Cantonment 47040, Pakistan; 2Department of Computer Engineering, HITEC University, Taxila Cantt, Rawalpindi 47080, Pakistan; 3Department of Mathematics, Lahore Campus, COMSATS University Islamabad, Lahore 54000, Pakistan; 4Department of Electrical Engineering, National University of Computer and Emerging Sciences, Islamabad 44000, Pakistan; 5Department of Computer Science and Engineering, Sejong University, Seoul 05006, Korea

**Keywords:** deep learning, machine learning, differential evolution, artificial bee colony, computer vision, skin-lesion segmentation

## Abstract

Rapid advancements and the escalating necessity of autonomous algorithms in medical imaging require efficient models to accomplish tasks such as segmentation and classification. However, there exists a significant dependency on the image quality of datasets when using these models. Appreciable improvements to enhance datasets for efficient image analysis have been noted in the past. In addition, deep learning and machine learning are vastly employed in this field. However, even after the advent of these advanced techniques, a significant space exists for new research. Recent research works indicate the vast applicability of preprocessing techniques in segmentation tasks. Contrast stretching is one of the preprocessing techniques used to enhance a region of interest. We propose a novel hybrid meta-heuristic preprocessor (DE-ABC), which optimises the decision variables used in the contrast-enhancement transformation function. We validated the efficiency of the preprocessor against some state-of-the-art segmentation algorithms. Publicly available skin-lesion datasets such as $PH^{2},$ ISIC-2016, ISIC-2017, and ISIC-2018 were employed. We used Jaccard and the dice coefficient as performance matrices; at the maximum, the proposed model improved the dice coefficient from 93.56% to 94.09%. Cross-comparisons of segmentation results with the original datasets versus the contrast-stretched datasets validate that DE-ABC enhances the efficiency of segmentation algorithms.

## 1. Introduction

Computer-aided design (CAD) technology has appeared widely in the medical imaging field. In addition, emerging deep-learning and machine-learning algorithms provide easy ways for researchers to create efficient and intelligent methods in this research field. Skin-lesion analysis is a significant subfield of medical imaging that helps diagnose and identify fast-spreading skin cancer. Therefore, it requires efficient methodologies for image analysis to further aid medical practitioners in expediting and enhancing the process of in-time and accurate identification of this ailment. Moreover, as a significant source of ultraviolet rays, the Sun may be the reason for this disease, due prolonged exposure to its rays striking the human body [1]. According to the World Health Organisation (WHO), cancer converts healthy body mass into tumour tissues in a multi-stage process in which cells may develop from premalignant to malignant cells. Thus, melanoma is the transformation of benign (non-cancerous) tissue to premalignant (having the potential to be malignant) tissue, and, finally, to malignant (cancerous) tissue. Therefore, melanoma is a menacing skin cancer and is the most important of its kind to accurately diagnose in time.

As per the American Cancer Society (ACS) statistics [2], more than six hundred thousand cancer patients are expected to die from cancer in 2022 in the US alone, approximately 1700 deaths per day. In addition, the expected number of new reported cases in the current year is around 1.9 million, out of which about one hundred thousand will be in situ melanoma. The report further states that it is the second deadliest disease among children aged 1 to 14 in the US. In 2022, the expected number of reported cancer cases will be above ten thousand, with one thousand above the death rate in the aforementioned age group [2].

Therefore, prompt diagnosis and proper classification of this ailment are of high priority. In skin-lesion classification, segmentation plays a preprocessing role in differentiating between the foreground and background of images, thereby pinpointing the region of interest (ROI). Researchers have proposed different skin-lesion segmentation techniques in the past, mainly divided into three broad categories: supervised, semi-supervised, and unsupervised [3]. In literature, skin-lesion segmentation is an effective technique. However, it is influenced by a number of variables such as varying brightness, contrast and shapes, artefacts such as hairs, skin types, and complex anatomical structures of the body. Due to this fact, an epidemiologist may be hindered in examining low-contrast images with noise [3]; this has led to the proposal of advanced segmentation methods, including deep-learning and machine-learning techniques.

Even after the proposal of advanced algorithms achieving improved outcomes, there exists a considerable probability of improvement in this field. The main aim of this study is to highlight the effect of contrast stretching on skin-lesion segmentation issues. We present differential evolution–artificial bee colony (DE-ABC) as a novel hybrid metaheuristic preprocessing method for contrast enhancement. We conducted exclusive experimentation and analysis on some publicly available skin-lesion datasets such as the International Skin Imaging Collaboration (ISIC) from 2016 [4], 2017 [5], and 2018 [6], and the dataset of Pedro Hispano hospital $\left({PH}^{2}\right)$ [7]. Skin-image samples from these datasets are depicted in Figure 1.

We propose an efficient brightness-preserving local area-based estimation algorithm for contrast enhancement to fill this gap. The main contributions of this paper are:A novel hybrid metaheuristic technique, DE-ABC, to estimate optimal parameters for local region-based brightness-preserving contrast stretching;Analysis of the effects of the proposed preprocessor on different state-of-the-art skin-lesion ROI segmentation models;Performance approximation of the proposed model as a medical imaging preprocessor using comparative study analysis.

The rest of the manuscript is organised as follows. First, a detailed literature review stating the gaps in the related works is discussed in Section 2. The later section discusses our proposed hybrid model, mathematical formulations, and parameter estimation. Next, we used some segmentation models to validate our model’s efficiency in Section 4; this section comprises a parameter discussion for the training and testing of these segmentation models and a detailed performance comparison of segmentation results based on visual and performance matrices.

## 2. Related Work

In the literature, there is a variety of research material related to the subject, such as the researchers who proposed a two-step contrast-stretching technique [8]. First, they performed global (top-hat maximising) and local (standard deviation model) pixel contrast enhancement; later, they reconstructed enhanced images by fusing both outcomes. Lesion maps were generated for segmentation purposes using the reconstructed images with superpixel estimation. They used a pre-trained ResNet50 model for feature extraction and the grasshopper algorithm for feature selection based on the estimation cost function. Improved classification results were computed using the naïve Bayes classifier. For experimentation, they used datasets such as $PH^{2}$, ISBI 2016, and HAM10000. In their work, some of the imbalances in skin lesions discussed earlier, such as low contrast and deformities, are resolved by [9]. The technique they used to determine the stated issue was the implementation of MASK RCNN based on the region proposal network (RPN); using an ROI arrangement (tiny features mapping), they generated a finished mask of a segmented image with loss function implementation. For classification, they used a 24-layer CNN model and showed accuracy improvement in the results. They used the same datasets as [8] in their research. The authors of [10] followed extensive preprocessing steps to obtain enhanced images for segmentation. A novel method for artefact removal was introduced in their work, called maximum gradient intensity (MGI). The measures included vertical, horizontal, and diagonal pixel-level intensity estimations to generate a hair mask; and hair-pixel replication using dilation methods. The team implemented histogram equalisation for general contrast enhancement. They used the histogram as a bi-modal method in Otsu thresholding for segmentation. Features such as asymmetry index, border irregularity, colour score and diameter (ABCD), grey level co-occurrence matrix (GLCM), and some local binarised patterns were extracted for classification. They used a supervised learning algorithm by feeding labels to an ANN, and utilised a feed-forward neural network to categorise benign tumours vs. melanoma. The image sets used in this work were the ISIC archives in combination with $PH^{2}$.

Another work with the exact same goals was presented in [11]. First, they preprocessed the input images for artefact removal using Dull Razor and Gaussian filtration to smooth the input. Then, the enhanced picture was reworked with k-means colour-based clustering for ROI retrieval. The research computed almost the same features as extracted in the other mentioned works. The research goal was achieved using the ISIC 2019 dataset with eight classifiable cancer classes utilising a multiclass support vector machine (MSVM)-based classifier. Another study used the hybrid combination of classifiers as an ensemble approach [12]. Using two skin datasets—$PH^{2}$ and Ganster—they implemented two main algorithms for two-class (structuring-based stacks) and three-class divisions using hierarchical structural-based stacks.

Moreover, they implemented morphology operations for preprocessing before segmentation tasks, such as squaring the binary pixels for contrast improvement, median filtering, and high pass filtration for artefact detection. Finally, Otsu thresholding and top-hat filtering eradicated the hairs after the inpainting model. Then, they performed all the operations separately on each channel—red, green, and blue (RGB)—of the image; then, the enhanced media were fused to form a final picture. A skin cancer detection system was presented in [13]; preparatory steps involve median filtration for noise removal and a morphological closing operation based on dilation and erosion. Contrast is improved by using the MATLAB built-in function “imadjust”. All the operations are performed on each RGB colour channel separately and fused after completing the respective step. Finally, the samples are segmented based on Otsu thresholding, with two distinct values for dark and light parts of the image.

Furthermore, subtraction of the two masks manages the extraction of the ROI. The researchers extensively exhibited research to extract ABCD features and their weight estimation for the total dermoscopy value (TDV). Based on the TDV, their work presents an excellent comparative analysis of the proposed system’s results versus ground truth. Research conducted by [14] added some new methods for class segregation using available datasets of skin-cancer images: $PH^{2}$ and ISBI 2016 and 2017. First, Laplace filtering is used in lesion local regions; later, the Laplacian pyramid estimation of the whole image enhances the image’s contrast. The authors achieved input augmentation by rotation at four distinct angles of 45°, 60°, 135°, and 270°. Subsequently, the transformation of the image into the hue, saturation, and value (HSV) colour space improves the lesion further. Then, an XOR operation of the contrast-stretched sample with Laplacian filtering is used to achieve the range of pixels segregating the foreground from the background. The authors performed segmentation by feeding these boundaries to a VGG19 pre-trained CNN model. Finally, feature extraction is carried out with the Inception V3 model and fusion using hamming distance (HD); entropy-based feature selection supports classification with improved accuracy on the said datasets. Low contrast and occlusions such as hair cause disturbances in the later processes of segmentation and classification. To address this issue, [15] uses a closing operation to remove artefacts. Unsharp masking and rescaled intensity were the two techniques used to stretch the image’s contrast. The researchers fed combined data of original samples and two different contrast-enhanced sets to a high-scaled supervised deep network for class segregation. Postprocessing methods for contour sifting were used to supplement the segmentation work. The datasets used in the experimentation process were the ISBI 2017 and $PH^{2}$.

Utilising the concept of local and global contrast enhancement, succeeded by Dull Razor for occlusion removal, researchers presented work including a proposal of two distinct segmentation methods [16]. They first generated a binary mask through contour segmentation based on local portions. Then, a second binary mask was estimated using the J-value of the colour-quantised image sample; identified regions of the same colour were fused before the estimation step. Fusion of both the binary masks preceded their postprocessing, which included edge and threshold adjustment and hole filling; this resulted in efficient segmented images. The datasets utilised in this research were the $PH^{2}$ and ISIC 2017.

Researchers have proposed a 2D/3D registration model based on spatial histograms to extract statistical features [17]. They presented weighted histogram gradients susceptible to rotation and scaling instead of translational movements. The authors tested the proposed model on X-ray and CT images. In another work, the authors presented contrast enhancement, image texture, and colour information improvement [18]. Furthermore, the enhanced datasets after preprocessing were utilised in segmentation and feature selection methods. They used Dull Razor to remove hair artefacts in the skin-lesion images, and achieved contrast stretching through the colour data and texture information.

Better segmentation results aid the structured extraction of inherent features in skin lesion images. However, noise, artefacts, and insufficient contrast heavily impact these intrinsic features. Commonly, preprocessors are utilised to mitigate these issues [3]. We have detailed some of the preprocessors used in recent research in Table 1.

The detailed literature review shows that in the medical-imaging field, the input data received might include noise; therefore, increased resemblance in the foreground and background may occur. In addition, instruction deficits and ROI merging affect class segregation and ROI extraction models. This is why preprocessing techniques play an essential role in these situations. Even though a wide range of algorithms exist to resolve this issue, most of them achieve the task by normalising the intensity levels of low-contrast images, thereby increasing the lesion size, weakening minute pieces of information near ROI, or disturbing the brightness levels. Therefore, using multiple strategies resulted in improved results on images, and the conditions given below have already been achieved:Enhance lesions with superior ROI;Reduce noise, artefacts, poor contrast;Equalise colour distribution in the whole image.

We aim to propose a model to generate efficient results when the above requirements are not met.

## 3. Proposed Methodology

Before discussing the proposed method and enhancement and transformation functions, we present some previous strategies used for contrast stretching. For image enhancement, some classical techniques such as transform, point, and spatial operations have been used [24]. In the transformation operation, various kinds of filtration, such as linear, root or homomorphic, have been used. These methods use zero memory operations depending on a single source pixel. The image inverse transformation yields the final results of enhancement in the domain of spatial frequency. Techniques such as thresholding, histogram adjustment, and contrast enhancement are considered to be point operations. The main discrepancy in these methods is that they globally transform the input images to generate enhanced images without considering that some areas might need distinct levels of contrast stretching. As a result, methods such as linear transformation and histogram adjustment have achieved greater popularity. Contrast corrections performed using linear transformation attempt to linearly stretch the desired range of grey levels. Conversely, histogram equalisation accomplishes contrast correction by spreading out the frequently occurring intensity value over the whole range of the image. Another operation for neighbourhood-based enhancement is spatial transformation, which includes median, low, high, and bandpass filtering and unsharp masking. The disadvantages of this technique are the unnecessary enhancement of noise present in images, and sometimes smoothing of the areas where sharpness is needed [25].

The proposed model is comprised of (1) skin-lesion downsampling and colour space conversion to hue, saturation, and intensity; (2) channel separation to apply the transformation on the intensity channel; and (3) estimation of distinct decision variables for the contrast-stretching transforming function. The flow of the proposed methodology is presented in Figure 2. We utilised four publicly available datasets for experimentation purposes. The original images are preprocessed and their contrast is stretched through the DE-ABC algorithm. These preprocessed images are used as input images for the selected segmentation algorithms. A cross-comparison of segmentation results is presented in the Section 4.

### 3.1. Cost and Enhancement Functions

We present a contrast-stretching preprocessor that utilises methods such as local area enhancements, including local mean and standard deviation, based on neighbourhood pixels instead of global improvements. The starting steps include converting the input image to HSI colour space, followed by channel separation. Finally, the proposed method is implemented on the intensity channel to convert it to an enhanced version with improved contrast. The following steps demonstrate the transformation function for contrast enhancement.
(1)$$\Xi^{*}=T_{x}\left(\Xi \right),\;\;\;\;\Xi ,\Xi^{*}\in R^{R,S}$$

Here, the transformation functions $T_{x}$ transform the intensity channel Ξ of the original image to an enhanced version $\Xi^{*}$, with dimensions of *R*-rows and *S*-columns. Undoubtedly, classical methods such as adaptive histogram equalisation give improved results, but they are computationally expensive, as they assign values according to the n-neighbouring pixels. In contrast, we utilised a less time-consuming approach presented in [24] based on statistical methods, specifying the improved intensity levels of pixels using the following function.
(2)$$g\left(i,j\right)=\left(\delta \frac{M}{\sigma (i,j)+\beta }\right)\times \left[f\left(i,j\right)-\gamma \times \mu \left(i,j\right)\right]+\mu {\left(i,j\right)}^{\alpha }$$
whereas *f(i, j)* is the original intensity level and *g(i, j)* is the enhanced version. *M* is the global mean, and *(i, j)* is the centre pixel on which the operation is performed. The locally estimated mean and standard deviation of the pixel under consideration are μ(i,j) and σ(i,j), estimated within the k×k local neighbouring pixels, as shown in Figure 3.

The original statistical method presented in [25] was modified in [24]; the authors added the non-zero β, and γ values; β allows zero standard deviation and γ aids to subtract a fraction of the mean from the centre pixel intensity value. The final term ensures the brightness-preserving smoothness effect. The local mean μ(i,j) and standard deviation σ(i,j) are the results of neighbouring pixel effects, and the decision variable presented below remains the same for the entire image. The values of these variables are estimated through our proposed hybrid meta-heuristic method. Pixel transformation is performed using Equation (Equation 2), as presented in Figure 3.

An automatic contrast enhancement technique demands auto-adjusting decision variables, based on which the enhancement function operates. This is where our proposed method’s efficacy is important. Main block diagram of the proposed algorithm is presented in Figure 4. The intensity pixels are first enhanced using the pixel enhancement function presented in Equation (Equation 2).
(3)$$O_{x}\left(\Xi^{*}\right)=\frac{log\left(log\left(E\left(\Xi^{*}\right)\right)\right)\times n_{edgels}\left(\Xi^{*}\right)\times H\left(\Xi^{*}\right)}{R\times S}$$

Later, objective functions must be presented to test the estimated decision variables. The basis of this function depends on many factors. For example, many sharp edges with high-intensity values correspond to a high-quality image [25]. With this knowledge, we used a mixed combination of the objective function, based on the summation of the number of edge pixels $n_{edgels}\left(\Xi^{*}\right)$ and their intensity values $E(\Xi^{*})$, as well as the entropy value $H(\Xi^{*})$ of the enhanced intensity channel, where R and S are the horizontal and vertical dimensions of the intensity image. The edge intensities and edges are detected using a Sobel edge filter. The double log minimises the effect of edge intensity, as high values may lead to improper contrast over-stretching [24]. $E(\Xi^{*})$ is represented in the form of the Sobel kernel in Equation (Equation 4). The horizontal and vertical gradients of the Sobel filter can be estimated through the following equations presented in [26].
(4)$$E\left(\Xi^{*}\right)=\sum_{i\in R}\sum_{j\in S}\sqrt{h_{\Xi }{\left(i,j\right)}^{2}+v_{\Xi }{\left(i,j\right)}^{2}}$$



$$h_{\Xi }\left(i,j\right)=g_{\Xi }\left(i+1,j+1\right)+2g_{\Xi }\left(i+1,j\right)+g_{\Xi }\left(i+1,j-1\right)-\;\;g_{\Xi }\left(i-1,j-1\right)-2g_{\Xi }\left(i-1,j\right)-g_{\Xi }\left(i-1,j+1\right)$$





$$v_{\Xi }\left(i,j\right)=g_{\Xi }\left(i+1,j+1\right)+2g_{\Xi }\left(i,j+1\right)+g_{\Xi }\left(i-1,j+1\right)-\;\;g_{\Xi }\left(i-1,j-1\right)-2g_{\Xi }\left(i,j-1\right)-g_{\Xi }\left(i+1,j-1\right)$$



### 3.2. Differential Evolution (DE)

The decision-variable estimation process is based on bounded search-space estimation. All the variables are specified within lower and upper bounds as *lb* and *ub*. DE, as the first step of meta-heuristic optimisation, requires population initialisation.

The whole population is evaluated using the cost function defined in Equation (Equation 2), generating a fitness vector. The random population initialisation is performed through the following function.
(5)$$P_{i,0}=lb+rand\left(0,1)\times (ub-lb\right),\;\;\;\;\;i=0,1,\cdots ,N_{p}$$


array of decision variables → 

α



β



γ



δ




The generated population comprises *N_p_* vectors of decision variables, where one vector consists of four variables. Each member of the population undergoes the mutation operation using the Equation (Equation 6) followed by recombination process as presented in Figure 5. Here, four random vectors participate, with one vector having the best fitness.
(6)$$M_{i,T}=P_{best}+F\left(P_{r1,T}-P_{r2,T}\right)+F\left(P_{r1,T}-P_{r2,T}\right)$$

The scaling factor *F* is a randomly generated number with the boundary conditions of [0, 1]. *T* is the number of iterations the DE will run, and the indices of the participating vectors are randomly chosen from the population, fulfilling the condition of r1≠r2≠r3≠r4≠i, where *i* is the index of the target vector on which the mutation is performed.

Both the target vector and the donor vector take part in the preceding operation of the binomial crossover. The selection of elements depends on specific criteria mentioned in the following equation. Prob is a randomly generated number in the range [0, 1], whereas ind is a random index within the range [1, $N_{p}$], which ensures that at least one element is chosen from the donor vector.
(7)$$C_{i,T}=\left\{\begin{array}{cc} P_{i,T} & i\neq ind\;\;\&\;\;r>prob\\ M_{i,T} & i=ind\;\;or\;\;r\leq prob \end{array}\right.$$

After traversing the whole population, following the mutation and crossover, the process of bounding and selection begins. All the decision variables are bounded with lower and upper bounds (*lb* and *ub*). In DE, there is greedy selection (survival of the fittest); either the parent or the child may survive based on the fitness score.
(8)$$P_{i,T+1}=\left\{\begin{array}{cc} P_{i,T} & \mathrm{if}\;O_{x}\left(\Xi^{P}\right)>O_{x}\left(\Xi^{C}\right)\\ C_{i,T} & \mathrm{otherwise} \end{array}\right.$$

$O_{x}\left(\Xi^{P}\right)$ is the cost of the image calculated using Equation (Equation 3); the image $\Xi^{P}$ is the enhanced version of the intensity channel using the parent decision-variable set. $O_{x}\left(\Xi^{C}\right)$ is the cost of the image calculated using Equation (Equation 3); the image $\Xi^{C}$ is the enhanced version of the intensity channel with the child decision-variable set. One iteration of the DE completes here, and is repeated for up to *T* iterations.

### 3.3. Artificial Bee Colony (ABC)

After the desired iterations, the decision-variable set in the population is used in the following meta-heuristic method: ABC. This method refines the variable set based on specific parameters. ABC was first proposed in the year 2005 by Dervis Karaboga. It was published in 2007 in the *Journal of Global Optimisation* [27].

Due to its inspiration from the social colonies of animals, it is called a swarm-based algorithm. Artificial bee colony promises more efficient outcomes than other swarm-based optimisation techniques, such as ant colonies, bird flocks, fish schools, and particle swarms. Two main characteristics of swarm intelligence are self-organisation and worker division. This self-organisation is based on local search data without relying on global results. The parts of this organisation are negative/positive feedback, changes in search responses, and numerous dealings with other workers. At the same time, the division of workers is the concurrent performance of specialised labour. Unlike DE, the cost of the objective function is not the fitness function here. To calculate the fitness, Equation (Equation 9) is used.
(9)$$fitness=\left\{\begin{array}{cc} \frac{1}{1+f} & \mathrm{if}\;f\geq 0\\ 1+|f| & \mathrm{otherwise} \end{array}\right.$$

Here, *f* is the objective function cost calculated using Equation (Equation 3). The objective function remains the same as mentioned in Equation (Equation 3) in the previous method of DE. As ABC is a swarm-based method, the decision-variable set is made up of food sources. In the case of the worker division, all the food sources should be exploited and randomly assigned with new members based on the greedy selection discussed in detail here. The exploitation of food sources is divided into three steps: the employed-bee phase, the onlooker-bee phase, and the scout-bee phase.
(10)$$X_{new}^{j}=X^{j}+\phi \left(x^{j}-x_{p}^{j}\right),\;\;\;\;\;i=1,2,\cdots D$$

The onlooker-bee phase starts with the mutation process presented in Equation (Equation 10). Unlike the mutation process in DE, only one decision variable out of the *D* variable set is selected for mutation from the food source $x^{j}$ to be exploited, where *D* in our case is four. The partner variable vector $x_{p}^{j}$ for mutation is chosen randomly. ϕ is a random number in the range [−1, 1]. After the mutation process of a single food source, its boundary condition is satisfied. The boundary condition for the decision variables is the same for DE and ABC (*lb* and *ub*), as explicitly discussed in the Section 4. The parent–child comparison and greedy selection are performed using the fitness value in Equation (Equation 11).
(11)$$P_{i,T+1}=\left\{\begin{array}{cc} P_{i,T} & \mathrm{if}\;fit_{new}<fit\\ C_{i,T} & \mathrm{otherwise} \end{array}\right.$$

Success will reset the trial vector value if we obtain a better child. A trial vector is used to track the failures that occurred for each food source. In case of failure, the value of the trial is incremented by one. Exhausting the limit of trials will lead to the scout-bee phase, which will be discussed later, after exploiting all the food sources once the employed-bee phase is finished for this iteration.

The next phase is the onlooker-bee phase; random decision variables are selected from the partner vectors and food sources to be exploited, as in the previous phase. The only difference is that whether exploitation takes place or not depends on the probability value. The probability value is calculated using Equation (Equation 12) based on the fitness of every food source and the maximum fitness in the population.
(12)$$Prob_{i,T}=0.9\times \left(\frac{fit_{i,T}}{max(fit)}\right)+0.1$$

It is then compared with a random number (0, 1). If the random number generated is less than the respective probability, all the steps involved in the food source exploitation, such as mutation, bounding, greedy selection, and updating the trial value, take place, as discussed in the previous phase. On the other hand, in the case of a more significant random number than the probability value, the trial value is left unchanged and the current bee exploits the next food source. The main difference between the previous and current phases is that every food source generates a new solution in the employed-bee phase. However, in the onlooker-bee phase, the food source may not develop new solutions depending on the probability value versus the randomly generated number.

As the limit of trials is reached, the food source corresponding to the maximum value of trials is replaced by a new variable set. This new food source is generated using the same formula for generating a new random population in Equation (Equation 5). The newly added food source is not compared with the existing food source, so there no greedy selection occurs in this phase. In the end, we reset the respective trial value. The population updates all the decision variables of the respective food sources along with their new fitness.

The best decision-variable set, known as the best solution, is extracted from this hybrid meta-heuristic method. It is then passed to the transformation function described in Equation (Equation 1) to enhance the contrast of the intensity channel. The intensity channel is then fused with the corrected hue and saturation channels. Finally, this combined HSI image is converted back to an RGB image, which is now ready to be used with segmentation algorithms. The following Section 4 discusses the effects of preprocessing on segmentation results.

## 4. Results

### 4.1. Parameters and Settings

We proposed a preprocessing contrast-stretching technique. The transformation function implemented in our algorithm is discussed in detail in the last section. The best solution set of decision variables (α,β,γ,δ) is used to transform the contrast of the intensity channel. In the Section 4, we first present the parameters used in our algorithm. The lower and upper bounds for these decision variables are an improved arrangement of boundaries defined in [24], i.e., lb=[0000.5], and ub=[1.60.50.81.5]. For experimentation purposes, we used four publicly available skin-lesion datasets. We divided these datasets into three independent testing and training sets. The first set contained a combination of 200 $PH^{2}$ and 900 ISIC-2016 training images for training. For testing, we used 379 images from the ISIC-2016 testing dataset. The second set was comprised of ISIC-2017, with 2600 images in total, consisting of 2000 training and 600 testing images. Finally, the last group of for training tasks 1 and 2 contained 2594 ISIC-2018 images; due to the missing masks of test images, we further divided the dataset into 2076 training images and 518 testing images. We used a single NVIDIA GeForce GTX 1660 Super GPU to train and test the segmentation models. The proposed DE-ABC was implemented in MATLAB 2022a. In contrast, we tested the proposed preprocessor’s efficacy using Python segmentation algorithms with the help of the original author’s contribution codes. We evaluated the performance of segmentation algorithms using two performance indicators, *Jaccard (IoU)* and *dice (F1)*. The formulation of these indicators is given in Equations (Equation 13) and (Equation 14), where, the *Intersection = TP*, *Union = TP + FP + FN*, *Recall* = $\frac{TP}{TP+FN}\;\;$, and *Precision* = $\frac{TP}{TP+FP}$. True Positive, True Negative, False Positive, and False Negative are denoted as *TP*, *TN*, *FP*, *FN*.
(13)$$Jaccard\left(IoU\right)=\frac{Intersection}{Union}$$
(14)$$Dice\left(F1\right)=2\times \frac{Precision\times Recall}{Precision+Recall}$$

### 4.2. Segmentation Models

To validate DE-ABC, we tested it as a preprocessor with two state-of-the-art segmentation algorithms. Both of the algorithms were implemented in Python.

#### 4.2.1. Boundary-Aware Transformer (BAT) Segmentation Model

The first segmentation model for validating our preprocessing model was the boundary-aware transformer (BAT) [28]. We utilised the training parameters presented in the original work on the segmentation algorithm. First, we resized all the images to 512 × 512 with a minibatch size of 8. Next, we used Imagenet pretraining for the network encoder and fine-tuned it for 250 epochs. With an insignificant reduction in loss of validation within 10 epochs, we brought the learning rate to 50%. The coding environment for the segmentation model was Python.

Table 2 compares performance matrices achieved for segmentation results using the original method versus the DE-ABC enhanced datasets. The tabular presentation of outcomes clearly reveals that our preprocessing significantly affects the performance of the segmentation results. In addition, the high-volume dataset presents better outcomes, as the training datasets were comparatively large with high diversity.

Figure 6 presents a visual comparison of segmentation results. For representation, we have depicted two samples of skin lesions from a pool of extensive segmentation results. The segmentation results of the original image (Ξ) have less relevance to the ground truth. In contrast, the enhanced images ($\Xi^{*}$) in both samples achieved the greatest relevance with the ground truth; this reveals the effectiveness of our proposed algorithm.

#### 4.2.2. Comprehensive Attention CNN (CA-Net) Segmentation Model

The next comparative model of segmentation we used was a comprehensive attention CNN (CA-Net), presented in [29]. We again utilised the same training parameters; the mini-batch size was 16 and the number of training epochs was 250. We used adaptive estimation (Adam) at the start with a learning rate of $0.1^{-3}$, and reduced it to half after 100 epochs. The rest of the parameters remained the same as proposed in [29].

The experimental results of Jaccard and F1 indices are presented in Table 3 for both of the groups. The comparative analysis shows that the proposed model also acted effectively with this segmentation model. The segmentation outcomes of the enhanced image with the proposed technique resulted in better performance indicators. The size of datasets again affected the performance, yielding a slightly increasing trend with an increase in the volume of the datasets.

As in the previous discussion, we present two samples from the pool of comparative results in Figure 7. The first sample shows better segmentation results. Comparatively, the ROI of the original image is not clear due to the low contrast and low brightness of the original image. In the second sample, the lesion is over-segmented in the case of the original image. When the same image is fed to the segmentation algorithm after preprocessing with DE-ABC, it yields a comparatively more suitable segmentation mask.

## 5. Conclusions

We have proposed a local-area contrast-correcting preprocessor in this research. Our model’s brightness-preserving transformation function pivots on local neighbourhood-based mean and standard deviation values. The meta-heuristic technique estimates the decision variables to optimise the image transformation results based on two models, DE and ABC. We tested our model with some segmentation algorithms to validate its performance. Images after DE-ABC preprocessing were superior in segmentation task results compared to segmentation with the original datasets. We incorporated four publicly available datasets into our research in three distinct groups for training and testing. The visual and tabular results validate the usefulness of our proposed model.

In the future, we plan to extend this to the classification of skin lesions. Furthermore, testing on massive skin-cancer datasets such as HAM10000 [30] and ISIC-2019 [31] may result in diverse results. Furthermore, we aim to implement our model in agriculture image datasets in addition to medical images. Finally, we expect to extend the system to feature-extraction and selection models.

## Figures and Tables

**Figure 1 diagnostics-12-02625-f001:**
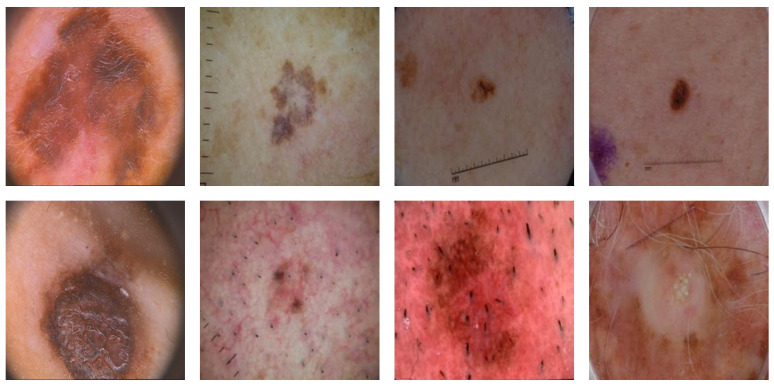
Two samples (**top** and **bottom**) of $PH^{2}$ and ISIS 2016, 2017, and 2018 (**left** to **right**).

**Figure 2 diagnostics-12-02625-f002:**
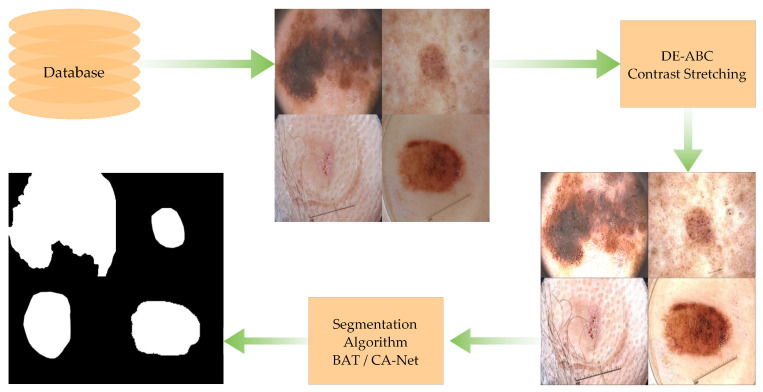
Proposed methodology general flow diagram.

**Figure 3 diagnostics-12-02625-f003:**
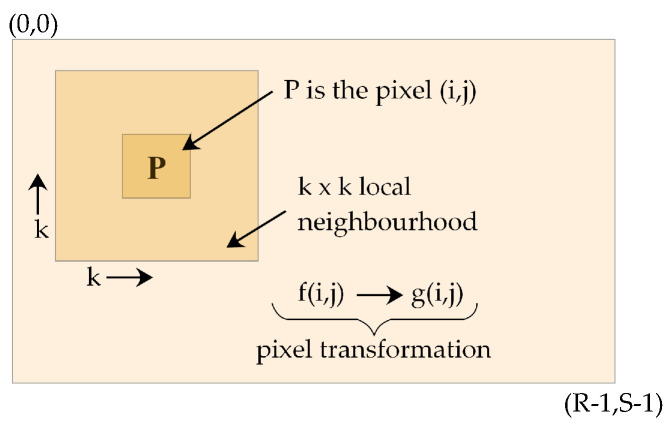
Pixelwise mask operation based on local neighbourhood.

**Figure 4 diagnostics-12-02625-f004:**
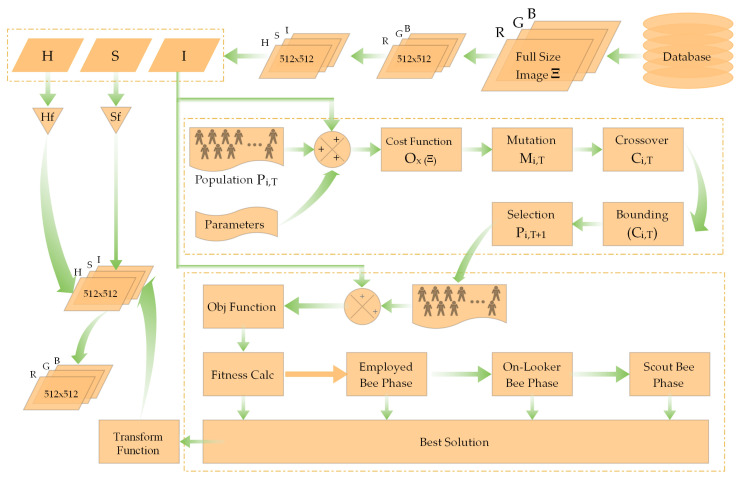
Block diagram of proposed methodology.

**Figure 5 diagnostics-12-02625-f005:**
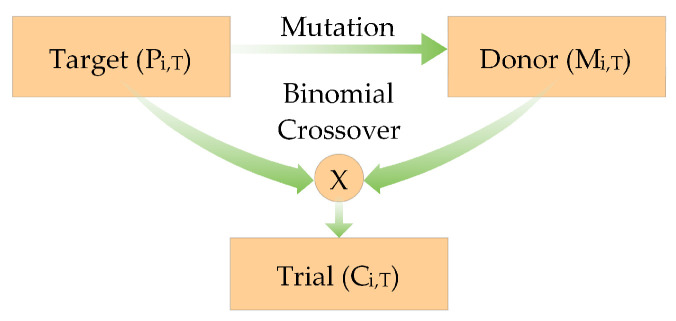
Each target vector undergoes the mutation, followed by the crossover operation.

**Figure 6 diagnostics-12-02625-f006:**
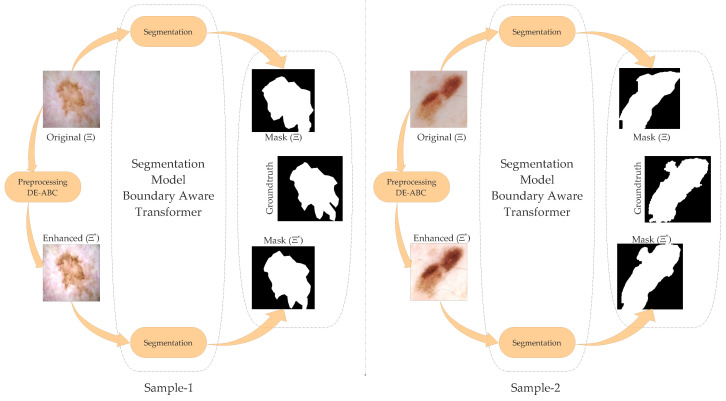
DE-ABC effects on boundary-aware transformer segmentation.

**Figure 7 diagnostics-12-02625-f007:**
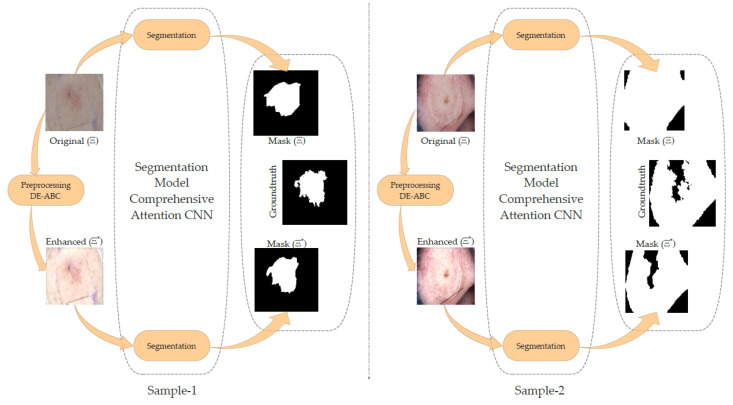
DE-ABC preprocessing effects on comprehensive attention CNN segmentation.

**Table 1 diagnostics-12-02625-t001:** Preprocessors utilised in segmentation process of research.

Study	Preprocessing Method
[19]	Contrast correction with equalised histogram and median-filtering utilisation to remove noise
[20]	Anisotropic diffusion, colour normalisation, and contrast stretching
[18]	Distinctive physical composition, i.e., texture data, is used for contrast stretching to approximate normalisation of luminance
[21]	Fast marching methods (FMM) utilised for hair removal and contrast stretching using contrast-limited adaptive histogram equalisation (CLAHE)
[22]	Edge enhancement and contrast correction performed using Canny, Sobel, and Gaussian filtration
[23]	Otsu thresholding utilised to correct the contrast of coloured images

**Table 2 diagnostics-12-02625-t002:** Experimental results of boundary-aware transformer (BAT) model.

Model	Database		Jaccard (IoU)	Dice (F1)
Boundary-aware transformer [28]	Original	ISIC 2016 + $PH^{2}$	0.8402	0.9123
		ISIC 2017	0.8593	0.9238
		ISIC 2018	0.8525	0.9198
	DE-ABC enhanced	ISIC 2016 + $PH^{2}$	0.8634	0.9267
		ISIC 2017	0.8693	0.9301
		ISIC 2018	0.8615	0.9256

**Table 3 diagnostics-12-02625-t003:** Experimental results of comprehensive attention CNN.

Model	Database		Jaccard (IoU)	Dice (F1)
Comprehensive attention CNN [29]	Original	ISIC 2016 + $PH^{2}$	0.8573	0.9232
		ISIC 2017	0.8741	0.9328
		ISIC 2018	0.8790	0.9356
	DE-ABC enhanced	ISIC 2016 + $PH^{2}$	0.8668	0.9287
		ISIC 2017	0.8864	0.9398
		ISIC 2018	0.8884	0.9409

## Data Availability

No new dataset was generated from this study. We utilised four publicly available datasets in this study. ISIC datasets URL: https://challenge.isic-archive.com/data/ (accessed on 27 July 2022), $PH^{2}$ dataset URL: https://www.fc.up.pt/addi/ph2%20database.html (accessed on 27 July 2022).

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
