# Peer review of "A Hybrid Preprocessor DE-ABC for Efficient Skin-Lesion Segmentation with Improved Contrast"

_diagnostics, 2022, doi:10.3390/diagnostics12112625_

Round 1

Reviewer 1 Report

In the current paper, the authors propose a hybrid preprocessor DE-ABC for efficient skin lesion segmentation with improved contrast. After reading the whole content, the study topic is interesting and the paper is well-written, here are some comments:

(1) In abstract, some numerical results can be listed to show the merits of the proposed method.

(2) It is unusual that Section 1 only contains one paragraph, thus the authors can reorganize the structure of Section by adding study gaps and motives.

(3) The motives of choosing artificial bee colony can be further explained in Section 3.3.

(4) How to choose the random number in terms of Formula 3.10 in the experimental analysis.

(5) In future study topics, more detailed illustrations can be stated in conclusions.

(6) More recently published papers can be incorporated in Literature review with an updated unified reference style.

Author Response

Oct. 20, 2022

The Editor-in-Chief,

Diagnostics,

MDPI.

Subject: Authors’ Response to Reviewers’ Comments.

Dear Editor,

On behalf of all the authors, I would like to thank you for a swift evaluation of our Manuscript ID: diagnostics-1982540, titled “A Hybrid Preprocessor DE-ABC for Efficient Skin Lesion Segmentation with Improved Contrast.”

            Following a thorough discussion, we have concluded that most of the issues raised by the honourable reviewers were genuine and should not have been there in the initially submitted manuscript; we apologise for those mistakes. We have done our best to address all the raised concerns in our revised version, and we hope that the reviewers will find the revised manuscript much better than the last time. Overall, we have seen the reviewers’ comments as positive, and we thank them for their encouragement and appreciation.

            In the response sheet, we tried to answer each objection raised by the reviewers and highlighted what we intended to do to revise our manuscript accordingly. Please see the attachment.   

We are confident that we have clarified all the doubts and objections raised by the honourable reviewers in our response above. We assure you that our revised manuscript has addressed the concerns mentioned above, and we hope it will come up to your expectations. We now look forward to your positive response henceforth.

Thanking you in anticipation.

Sincerely yours,

Imran Ashraf, PhD (Corresponding author)

Assistant Professor,

Department of Computer Engineering,

HITEC University,

Pakistan.

Reviewer 2 Report

This work is not enough contribution and innovation. However, the problem statement and motivation could be stronger or more clearly highlighted.                                  

1.      The existing literature should be classified and systematically reviewed, instead of being independently introduced one-by-one.

2.      The abstract is too general and not prepared objectively. It should briefly highlight the paper's novelty as what is the main problem, how has it been resolved and where the novelty lies?

3.      For better readability, the authors may expand the abbreviations at every first occurrence.

4.      The author should provide only relevant information related to this paper and reserve more space for the proposed framework.

5.      However, the author should compare the proposed algorithm with other recent works or provide a discussion. Otherwise, it's hard for the reader to identify the novelty and contribution of this work.

6.      The descriptions given in this proposed scheme are not sufficient that this manuscript only adopted a variety of existing methods to complete the experiment where there are no strong hypothesis and methodical theoretical arguments. Therefore, the reviewer considers that this paper needs more works.

The algorithm presented has not any novelty.

7.      The related works section is very short and no benefits from it. I suggest increasing the number of studies and add a new discussion there to show the advantage. Following studies can be added:

a.      2D/3D Multimode Medical Image Alignment Based on Spatial Histograms.

b.      A Collaborative Alignment Framework of Transferable Knowledge Extraction for Unsupervised Domain Adaptation.

c.      A lightweight approach for skin lesion detection through optimal features fusion

d.      An improved strategy for skin lesion detection and classification using uniform segmentation and feature selection based approach

 The manuscript is not well organized. The introduction section must introduce the status and motivation of this work and summarize with a paragraph about this paper.

Author Response

(The authors gave the same response as above.)

Reviewer 3 Report

1.      Line 152, “Analyse” should be Analyze

2.      Please briefly introduce each element in equations 3.1 and 3.6.

3.      What is the “Decision Variables -> α β γ δ between line 205 and 206? Is that an equation? A table? Or A figure? Please assign an appropriate caption to it.

4.      Figure 4s resolution is low, please improve it.

5.      In line323 and equation 4.1, the letter U's font format in IoU and Union seems not right, please correct it.

6.      In equation 4.2, the authors use Dice (Sørensen), but in table 1 and table2, they use Dice (F1). Although they are the same equation, please keep it consistent.

7.      How long will the proposed method take to process one image?

Author Response

(The authors gave the same response as above.)

Round 2

Reviewer 2 Report

The paper is improved and can be accepted